

# Classification of imbalanced ECGs through segmentation models and augmented by conditional diffusion model

Jinhee Kwak and Jaehee Jung

Department of Information and Communication Engineering, Myongji University, Yongin, Gyeonggi-do, Republic of South Korea

Corresponding author
Jaehee Jung, jhjung@mju.ac.kr

## ABSTRACT

Electrocardiograms (ECGs) provide essential data for diagnosing arrhythmias, which can potentially cause serious health complications. Early detection through continuous monitoring is crucial for timely intervention. The Massachusetts Institute of Technology-Beth Israel Hospital (MIT-BIH) arrhythmia dataset employed for arrhythmia analysis research comprises imbalanced data. It is necessary to create a robust model independent of data imbalances to classify arrhythmias accurately. To mitigate the pronounced class imbalance in the MIT-BIH arrhythmia dataset, this study employs advanced augmentation techniques, specifically variational autoencoder (VAE) and conditional diffusion, to augment the dataset. Furthermore, accurately segmenting the continuous heartbeat dataset into individual heartbeats is crucial for confidently detecting arrhythmias. This research compared a model that employed annotation-based segmentation, utilizing R-peak labels, and a model that utilized an automated segmentation method based on a deep learning model to segment heartbeats. In our experiments, the proposed model, utilizing MobileNetV2 along with annotation-based segmentation and conditional diffusion augmentation to address minority class, demonstrated a notable 1.23% improvement in the F1 score and 1.73% in the precision, compared to the model classifying arrhythmia classes with the original imbalanced dataset. This research presents a model that accurately classifies a wide range of arrhythmias, including minority classes, moving beyond the previously limited arrhythmia classification models. It can serve as a basis for better data utilization and model performance improvement in arrhythmia diagnosis and medical service research. These achievements enhance the applicability in the medical field and contribute to improving the quality of healthcare services by providing more sophisticated and reliable diagnostic tools.

## INTRODUCTION

Arrhythmia refers to an irregular heartbeat resulting from abnormal electrical signals, and its diagnosis is typically conducted through an electrocardiogram (ECG). ECG provides a temporal sequence of graphical depictions illustrating the heart's electrical

activity. Regular monitoring of the ECG is imperative for managing arrhythmia since the symptoms are often transient or asymptomatic. ECG can be measured in the hospital and with portable devices for everyday use. Remarkably, hospitals commonly employ 12-lead, whereas wearable devices typically utilize a single lead, a practice supported by several studies (*Saadatnejad, Oveisi & Hashemi, 2019*; *Wang et al., 2019*; *Meng et al., 2021*), which has demonstrated a notable reduction in measurement discomfort. Therefore, the critical need lies in establishing dependable and enduring ECG monitoring capabilities and algorithms capable of accurately detecting arrhythmias through wearable devices in everyday life. Consequently, ongoing research is focused on categorizing various forms of arrhythmias from ECGs (*Manisha, Dhull & Singh, 2020*; *Mathunjwa et al., 2021*; *Malik et al., 2021*; *Mathunjwa et al., 2022*; *Wang et al., 2022*; *Zahid, Kiranyaz & Gabbouj, 2022*; *Ivora et al., 2022*).

Arrhythmias encompass a diverse range of types, with patterns exhibiting individual variations. ECG data represents a distinctive PQRST pattern, and the classification of arrhythmias is derived by analyzing the phase and amplitude of each waveform within PQRST pattern. Consequently, the technical pattern analysis of arrhythmia using ECG data serves not only to assist medical professionals in clinical arrhythmia assessment but also enables self-diagnosis through automated mechanical judgment.

The Massachusetts Institute of Technology-Beth Israel Hospital (MIT-BIH) arrhythmia dataset (*Goldberger et al., 2000*), widely employed in numerous arrhythmia classification studies (*Manisha, Dhull & Singh, 2020*; *Mathunjwa et al., 2021*; *Malik et al., 2021*; *Mathunjwa et al., 2022*; *Wang et al., 2022*; *Zahid, Kiranyaz & Gabbouj, 2022*; *Ivora et al., 2022*), exhibits an inherent imbalance among its arrhythmia classes. This is because the incidence of arrhythmias varies, and the data is based on a small number of patients. Table 1 illustrates the categorization options based on the standard of the Association for the Advancement of Medical Instrumentation (AAMI) (*Association for the Advancement of Medical Instrumentation, 1998*). The AAMI classification can further be subcategorized by utilizing the MIT-BIH symbol. In the MIT-BIH arrhythmia dataset, adherence to AAMI standard results in the grouping of 15 arrhythmias into N, S, V, F, and Q classes. N class represents a regular beats with abnormalities in the PQRST waves. S class denotes an arrhythmia characterized by premature or ectopic beat in the atria, while V class involves premature or ectopic ventricle beats. F class is a fusion of ventricular and normal beats, and Q class encompasses paced beats influenced by a pacemaker, coordinating atrial and ventricular beats in patients with a slower-than-normal pulse and unclassifiable beats.

*Ochiai, Takahashi & Fukazawa (2018)* employed data rates for the N, S, V, F, and Q classes, with respective values of 89.46%, 2.74%, 7%, 0.79%, and 0.01%. The imbalance in data distribution across classes is primarily attributed to the prevalence of normal beats within N class, despite varying incidence rates. As arrhythmia manifests as irregularities following normal beats, the overrepresentation of normal beats is unavoidable. Therefore, *Ochiai, Takahashi & Fukazawa (2018)* analyzed that imbalance in arrhythmia classes can potentially reduce the performance of classification models.

Instead of classification according to the AAMI standard, many studies have presented classification models using MIT-BIH symbols that can well identify the pattern of each

**Table 1  The correspondence between AAMI classes and MIT-BIH arrhythmia classes.**

| AAMI | | MIT-BIH | |
|---|---|---|---|
| Symbols | Classes | Symbols | Classes |
| N | Non ectopic beat | N | Normal beat |
| | | L | Left bundle branch block beat (LBBB) |
| | | R | Right bundle branch block beat (RBBB) |
| | | e | Atrial escape beat |
| | | j | Nodal (junctional) escape beat |
| S | Supraventricular ectopic beat | A | Atrial premature beat |
| | | a | Aberrated atrial premature beat |
| | | J | Nodal (junctional) premature beat |
| | | S | Supraventricular premature beat |
| V | Ventricular ectopic beat | V | Premature ventricular contraction |
| | | E | Ventricular escape beat |
| F | Fusion beat | F | Fusion of ventricular and normal beat |
| Q | Unclassifiable beat | / | Paced beat |
| | | f | Fusion of paced and normal beat |
| | | Q | Unclassifiable beat |

waveform (*Faziludeen & Sabiq, 2013*; *Cho et al., 2015*; *Rana & Kim, 2019*; *Yang & Wei, 2020*). Particularly in models designed for the automated classification of individual heartbeat waveforms, *Ji, Zhang & Xiao (2019)* proposed a classification model highlighting normal, L, and R symbols, chosen for their relatively abundant data among various symbols in the MIT-BIH dataset.

Therefore, this article aims to improve classification performance across all classes of arrhythmias, mitigate data imbalance issues, and address segmentation and classification within a single framework using a deep learning model. The following points are emphasized through this article.

- By classifying arrhythmias using the comprehensive AAMI classification system rather than the narrower MIT-BIH subclassification scheme, the arrhythmia classification model can be generalized.
- Due to the nature of medical data, there is an imbalance in the data for each classification. Therefore, this article proposes a model that shows balanced performance across all classes by comparing arrhythmia classification models that apply augmentation techniques using machine learning and deep learning.
- The performance of arrhythmia classification is compared by applying the method used R-peak label to segment the heartbeat into one unit and the method that automatically segments the heartbeat through waveform analysis. The automated segmentation model is a straightforward end-to-end model that performs segmentation and classification with a single network. Therefore, this model can help improve performance by sharing the same features across multiple stages.

## RELATED WORK

### ECG classification analysis

Studies on arrhythmia classification through ECG analysis algorithms can be categorized into two main groups: methods based on pattern recognition of PQRST waves and methods employing deep learning models. Table 2 represents the arrhythmia classification models within these two primaries groups.

### *PQRST-waves pattern recognition model*

The continuous MIT-BIH dataset composed of multiple heartbeats is segmented into individual heartbeats for arrhythmia classification purposes. These segmented waveforms consist of PQRST waves, with the features of each waveform being utilized for diagnosing arrhythmias. In Table 2, *Faziludeen & Sabiq (2013)* utilized the Pan-Tompkins algorithm (*Pan & Tompkins, 1985*) to detect the QRS complex. Subsequently, 25 features were extracted from the QRS complex by applying the Daubechies four wavelet, which included the mean, variance, standard deviation, the maximum and minimum of the detail coefficient, as well as approximation coefficient. Additionally, the R-R interval (RRI) was employed as a feature for classifying normal, LBBB (L), and PVC (V) using a support vector machine (SVM), achieving an accuracy of 98.95% (*Faziludeen & Sabiq, 2013*). *Senapati, Senapati & Maka (2014)* conducted classification of five arrhythmia classes using the scaled conjugate gradient (SCG) algorithm, employing RRI, QRS interval, and morphological features of QRS and T waves as input. The achieved accuracy was 96.2%, with a sensitivity of 96.5%. However, it was noted that the classification performance for the S class was lower compared to other classes due to the limited amount of train data. In *Cho et al. (2015)*, nine patterns were defined based on the amplitude and phase of the QRS. By employing these nine patterns to classify six arrhythmias, a detection rate of 93.72% was achieved. *Li & Zhou (2016)* employed wavelet packet entropy (WPE) of the QRS complex and the RRI as features. Random forest (RF) was then utilized for the classification of N, S, V, F, and Q classes, achieving an accuracy of 94.61%. Additionally, *Yang & Wei (2020)* utilized the Pan-Tompkins algorithm to extract the QRS complex. To identify the P wave, the QRS complex was utilized, and features were extracted from the maximum amplitude. Additionally, the visual morphological pattern of QRS complex (VMP-QRS) algorithm was employed to extract visual features from the data, while the adaptive K-means clustering (AKMC) algorithm was utilized to analyze the features of the data through clustering. Various models including neural networks, radial basis function based support vector machine (RBF-SVM), and K-nearest neighbor (KNN) were employed for the classification of the fifteen arrhythmias. *Yang & Wei (2020)* noted that the proposed models demonstrated improved performance in automated arrhythmia classification techniques, although their emphasis on visual patterns posed a limitation. It is worth mentioning that most of the research articles on pattern recognition for arrhythmia classification predominantly focus on morphological patterns.

**Table 2  Arrhythmia classification models using ECG data.**

| Method | Reference | Features | Class | | Classifier | ACC (%) |
|---|---|---|---|---|---|---|
| | | | **AAMI Class** | **MIT-BIH Class** | | |
| Pattern Recognition | *Faziludeen & Sabiq, 2013* | Features of wavelets of each beat (mean, variance, standard deviation, minimum and maximum of detail coefficients and of approximation coefficients), RRI | – | Normal, L, V | SVM | 98.95 |
| | *Senapati, Senapati & Maka, 2014* | RRI, QRS Interval, QRS & T-wave morphology | N, S, V, F, Q | – | SCG | 96.20 |
| | *Cho et al., 2015* | $Q, \overline{Q}, R, r, \overline{R}, \acute{R}, S, s$ | – | Normal, V, A, L, R, Paced beat | $QRs, QRS, RS, rS, R, r, \overline{R}$ or $\overline{r}, \overline{Q}\,\overline{R}, R\,\acute{R}$ | 93.72 |
| | *Li & Zhou, 2016* | QRS Wavelet packet entropy, RRI | N, S, V, F, Q | – | RF | 94.61 |
| | *Yang & Wei, 2020* | Amplitude, Interval, Duration, Visual morphological pattern-QRS | – | Normal, L, R, e, j, A, a, J, S, V, E, F, P, f, Q | KNN | 97.70 |
| Deep Learning | *Ochiai, Takahashi & Fukazawa, 2018* | – | S, V | – | CDAE | 96.85 |
| | *Rana & Kim, 2019* | – | – | Normal, L, R, A, V | LSTM | 95.00 |
| | *Xu, Jeong & Li, 2020* | – | N, S, V, F, Q | – | CNN +BiLSTM | 95.90 |
| | *Qiu et al., 2021* | – | N, S, V, F | – | Faster R-CNN | 95.68 |
| | *Anis & Sharma, 2022* | – | N, S, V, F, Q | – | CNN | 96.60 |

### Deep learning model

Classifying arrhythmias through pattern recognition poses challenges due to the substantial differences in ECG signals among patients. Consequently, it has become commonplace to employ deep learning models for feature extraction (*Liu et al., 2021*). *Ochiai, Takahashi & Fukazawa (2018)* utilized a convolutional denoising autoencoder (CDAE) model, incorporating a convolution layer into a denoising autoencoder (DAE), for the classification of two arrhythmia classes. However, the outcomes revealed a notably lower sensitivity and positive predictive value for the S class. The reason for this limitation was that the number of training data for S class was relatively small compared to other classification classes. *Rana & Kim (2019)* proposed a long short term memory (LSTM) model to classify the normal, LBBB (L), RBBB (R), Atrial Premature (A), and PVC (V) classes with 95% accuracy. *Xu, Jeong & Li (2020)* employed a transfer-learned convolutional neural network+bidirectional long short term memory (CNN+BiLSTM) classification model with the 2017 PhysioNet/CinC Challenge dataset. The model achieved an accuracy of 95.9% in classifying five AAMI arrhythmia classes: N, S, V, F, and Q. *Qiu et al. (2021)* employed the synthetic minority oversampling technique (SMOTE) to augment the data and showed improved performance in arrhythmia classification using the Faster R-CNN model. *Anis & Sharma (2022)* demonstrated that the CNN model surpassed previous techniques in terms of classification accuracy and efficiency for N, S, V, F, and Q classes.

Comparing the articles that classified AAMI arrhythmia classes in Table 2, it can see that the accuracy of arrhythmia classification models using deep learning is relatively high compared to the accuracy of pattern recognition. Therefore, the performance of deep learning arrhythmia classification methods is better than that of pattern recognition methods. This study evaluates the performance of the ECG classification model against two models, MobileNetV2 and the transformer, which have demonstrated strong performance among various deep learning architectures.

MobileNetV2 (*Sandler et al., 2018*), introduced by Google, represents a lightweight residual CNN architecture predominantly utilized in mobile and embedded systems. Its distinction from traditional CNN architectures lies in the adoption of depthwise separable convolutions, enabling the computation of convolutions with a substantially reduced parameter count. This method involves executing depthwise convolutions, which apply a filter to each input channel for capturing spatial features, followed by pointwise convolutions that aggregate the depthwise convolution outputs through 1x1 convolutions. Furthermore, MobileNetV2 innovates with the introduction of the inverted residual structure, which serves to diminish computational load while improving accuracy. This structure is composed of two block types: those with a stride of 1, mirroring residual blocks in preserving feature map dimensions and facilitating learning, and those with a stride of 2, which halve the feature map size, aiding in the model's efficiency. Consequently, MobileNetV2, despite its efficiency-oriented design, achieves notable performance.

The transformer model, which overcomes the long-term dependency issues inherent in recurrent neural networks (RNNs), is a Sequence-to-Sequence model that has served as the foundation for revolutionary advances in the field of natural language processing. It is composed of encoders and decoders that utilize the attention mechanism (*Vaswani et al.,*

*2017*). The attention mechanism calculates the importance of different positions from each position in the input, considering their relationships. The calculation process of attention is as given in Eq. (1).

$$\text{Attention}(Q, K, V) = \text{softmax}(\frac{QK^T}{\sqrt{d_k}})V. \tag{1}$$

The symbols $Q$, $K$, and $V$ denote the constructs of query, key, and value, respectively, with each being matrices derived from the multiplication of the input matrix X by the corresponding weight matrices $W_i^Q, W_i^K, W_i^V$. The $d_k$ refers to the dimension of the key vector. Calculation of the attention score is achieved through the dot product between the Q and K matrices, followed by division by $\sqrt{d_k}$ to facilitate stable gradient computation. The attention distribution is then determined using the softmax function, and this distribution is subsequently applied to the V matrix to generate the attention matrix. Through the attention matrix, features from positions of highest importance at each location of the same input can be extracted. The transformer model can be utilized as a classifier by adding a linear layer to the trained encoder. Consequently, the employment of the attention mechanism enables the transformer to efficiently capture dependencies across all positions, with the benefit of parallel computation facilitating a substantial reduction in training.

## Data augmentation

Data augmentation is a technique that generates new data by reflecting the characteristics of the original dataset. In case of medical data, collecting data for training deep learning models is challenging due to issues such as privacy concerns and variations in disease incidence rates. Moreover, class imbalance can compromise the performance of classification models and lead to overfitting problems in deep learning models. Consequently, data augmentation methods are being developed to mitigate these challenges (*Wen et al., 2021*). For the data used in this article, the N class constitutes 56% of the entire MIT-BIH dataset, in contrast to the S, V, and F classes, which represent 12%, 29%, and 3%, respectively. Minority classes, such as the F class, which only comprise 3% of the dataset, can impact classification performance. Therefore, this manuscript utilizes three augmentation techniques, SMOTE-Tomek, VAE, and conditional diffusion, to adjust the quantities of the S, V, and F class data to match that of the N class, thereby achieving dataset balance.

Among the many methods, the reasons for using the three augmentation techniques, SMOTE-Tomek, VAE, and conditional diffusion, for the following reasons. SMOTE-Tomek combines the oversampling technique of SMOTE with the undersampling technique of Tomek-links. After generating data with SMOTE, Tomek-links is used to remove data, achieving class balance. VAE is a deep learning augmentation technique that generates data by training the latent distribution of the data. Therefore, it can augment similar data by extracting the features of the latent vectors. Conditional diffusion, an image generative model, trains the features of the data by adding noise to the data and then reconstructing the original data in the process.

### SMOTE-Tomek

SMOTE-Tomek is an augmentation method that employs both SMOTE (*Chawla et al., 2002*) oversampling and Tomek-links undersampling techniques. SMOTE generates data for the minority classes by utilizing the nearest neighbor algorithm. On the other hand, Tomek-links are used to remove data from the majority class that exists in pairs at the boundary between the majority and minority classes. Consequently, SMOTE-Tomek adjusts class balance by generating data with SMOTE and removing data using Tomek-links.

SMOTE was applied to the MIT-BIH arrhythmia dataset by *Pandey & Janghel (2019)* to perform oversampling. Subsequently, a classification of five arrhythmia classes was conducted using an 11-layer deep CNN model. The accuracy of the classification model trained on the augmented data showed an improvement of 2.26% compared to the CNN classifier trained on the original dataset. This indicates that using data augmented by SMOTE can enhance the model's classification performance. The MIT-BIH arrhythmia dataset was augmented using SMOTE by *Xiaolin, Cardiff & John (2020)*, followed by the classification of N, S, V, F, Q classes using a one-dimension CNN. Compared to the classifier trained on the imbalanced dataset, the performance of the classifier trained on the balanced dataset exhibited a decrease in accuracy by 1%, specificity by 1.54%, precision by 6.89%, and F1 score by 2.48%. However, an improvement of 2.09% was observed in sensitivity.

### VAE

VAE (*Kingma & Welling, 2022*) is a model designed to learn the distribution of data in order to generate similar data. It comprises an encoder and a decoder. The structure of the VAE utilized in this study is depicted in Fig. 1. In this study, the input to the VAE is a segment of a one-dimension ECG signal. This representation has the dimension of (500,1), since the signal has been segmented to represent a single heartbeat. In a VAE, the encoder plays a crucial role in estimating the distribution of the data. It estimates a latent vector containing the mean ($\mu$) and variance ($\sigma^2$) parameters of the input data distribution. These parameters are assumed to follow a Gaussian distribution. The decoder in a VAE uses latent vectors to learn to approximate the distribution of the data while reconstructing to the input data. The encoder in a VAE employs convolution to train the distribution of the input data. Consequently, the encoder approximates the distribution of the latent vector, denoted as **z**, given the input data **x**. This latent vector **z** represents the mean and variance of the input data. Subsequently, the decoder utilizes the latent vector and convolution to reconstruct the input data. It is important to note that the decoder is symmetric to the encoder in the VAE architecture. The loss function is equal to Eq. (2).

$$\text{Loss} = \mathbb{E}_{q_\phi(\mathbf{z}|\mathbf{x})}[\log(p(\mathbf{x}|\mathbf{z}))] - \text{KL}(q_\phi(\mathbf{z}|\mathbf{x})||p(\mathbf{z})). \tag{2}$$

The first term corresponds to the reconstruction loss, while the second term corresponds to the Kullback–Leibler Divergence (KLD). The reconstruction loss measures the difference between the input data and the reconstructed data. KLD is a function that approximates the probability distribution of an idealized latent variable and the probability distribution

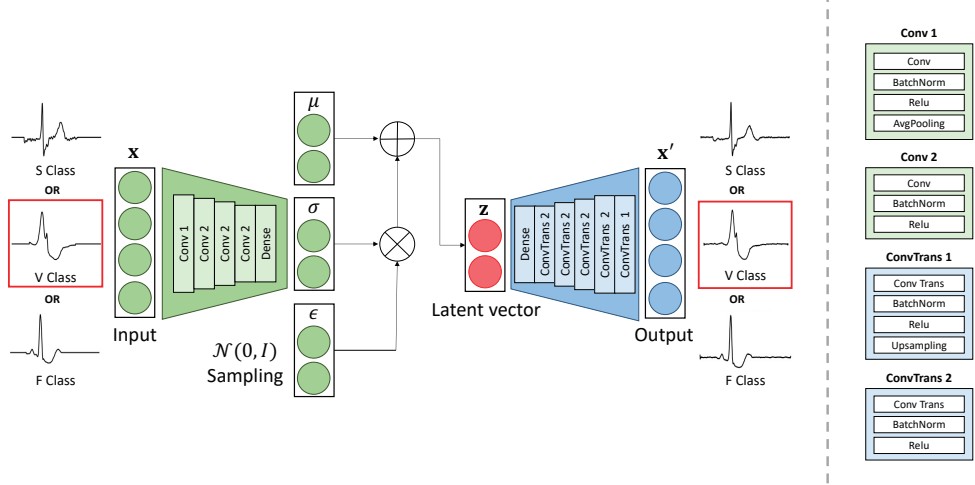

**Figure 1** The architecture of the proposed VAE model.

of an estimated latent variable. Consequently, during training, the model is optimized towards minimizing the overall loss function, which encompasses both the reconstruction loss and the KLD. This optimization process ensures that the model trains to accurately reconstruct the input data while also appropriately modeling the latent space distribution.

To evaluate the performance of the VAE model, the test data, 10% of the total dataset, was used to calculate the average cosine similarity per class. The test data was fed into the trained VAE model, and the decoder was used to reconstruct the data. Subsequently, the cosine similarity between the reconstructed data and the original data was then calculated. The obtained cosine similarity values were 0.99 for the S class, 0.99 for the V class, and 0.99 for the F class. Higher cosine similarity values indicate greater similarity between the original and reconstructed data, suggesting that the VAE model effectively estimated the distribution of the data.

*Kuznetsov et al. (2021)* used VAE in the Lobachevsky University Electrocardiography Database (LUDB) to generate an ECG for a single heartbeat. *Kuznetsov et al. (2021)* presented a VAE model that outputs two 25-dimension vectors from the encoder and inputs them to the decoder. The maximum mean discrepancy value used as an evaluation metric for VAE is $3.83 \times 10^{-3}$. *Kuznetsov et al. (2021)* stated that their proposed model is relatively lightweight, but it cannot generate the entire ECG sequence.

### Conditional diffusion

Diffusion (*Ho, Jain & Abbeel, 2020*) is a generative model that trains the data distribution through a forward process, adding Gaussian noise to the initial data $\mathbf{x}_0$, and a reverse process, eliminating the noise from the Gaussian-noised data $\mathbf{x}_T$, thus reconstructing it to the original data $\mathbf{x}_0$, as illustrated in Fig. 2. Given that the forward and reverse processes

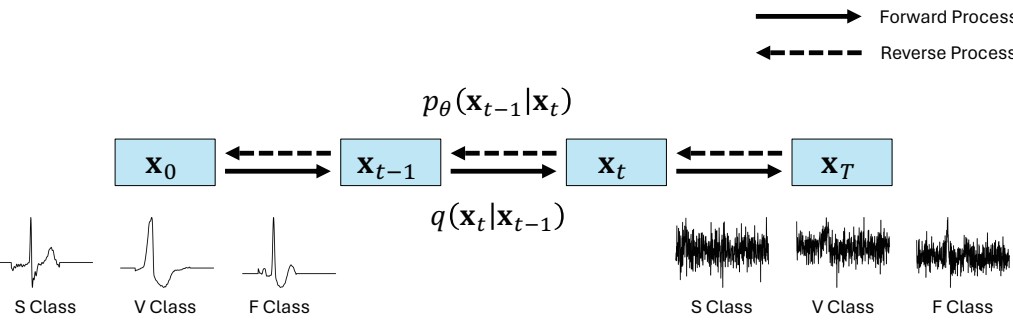

**Figure 2** **Conditional diffusion model for ECG synthesis.**

are defined as a Markov chain, there exists the capacity to progressively add and remove noise from the data.

Considering the data $\mathbf{x}_{t-1}$ in the forward process, Eq. (3) illustrates the procedure of adding Gaussian noise and subsequently computing the data $\mathbf{x}_t$ at the timestamp. In Eq. (3), $\beta_t$ is the variance schedule. Using $\beta_t$ in Eq. (3), defining $\alpha_t = 1 - \beta_t, \bar{\alpha}_t = \Pi_{i=1}^t \beta_t$, Eq. (3) can be expressed as Eq. (4). Equation (4) signifies the ability to directly compute $\mathbf{x}_t$, by adding Gaussian noise to the original data $\mathbf{x}_0$. Through iterative application of the reparameterization trick to Eq. (4), the noise can be calculated, as demonstrated in Eq. (5).

$$q(\mathbf{x}_t | \mathbf{x}_{t-1}) := \mathcal{N}(\mathbf{x}_t; \sqrt{1 - \beta_t} \mathbf{x}_{t-1}, \beta_t \mathbf{I}) \tag{3}$$

$$q(\mathbf{x}_t | \mathbf{x}_0) = \mathcal{N}(\mathbf{x}_t; \sqrt{\bar{\alpha}_t} \mathbf{x}_0, (1 - \bar{\alpha}_t) \mathbf{I}) \tag{4}$$

$$\mathbf{x}_t = \sqrt{\bar{\alpha}_t} \mathbf{x}_0 + \sqrt{1 - \bar{\alpha}_t} \epsilon \quad \epsilon \sim \mathcal{N}(0, \mathbf{I}). \tag{5}$$

In the reverse process, the process of reconstructing the original data $\mathbf{x}_0$ with noise $\mathbf{x}_t$ that follows a Gaussian distribution is shown in Eq. (6).

$$p_\theta(\mathbf{x}_{t-1} | \mathbf{x}_t) := \mathcal{N}(\mathbf{x}_{t-1}; \mu_\theta(\mathbf{x}_t, t), \Sigma_\theta(\mathbf{x}_t, t)). \tag{6}$$

As the reverse process requires training, the loss function is equivalent to Eq. (7).

$$L_{\text{simple}}(\theta) := \mathbb{E}_{t, \mathbf{x}_0, \epsilon}[||\epsilon - \epsilon_\theta(\sqrt{\bar{\alpha}_t} \mathbf{x}_0 + \sqrt{1 - \bar{\alpha}_t} \epsilon, t)||^2]. \tag{7}$$

Equation (7) stands for finding the minimizing KLD between the the probability distribution of the actual reverse process and the probability distribution of the denoised $\mathbf{x}_{t-1}$ given $\mathbf{x}_t$ as input to the U-Net model.

In this study, one-dimension ECG sequence data was reshaped into two-dimensions and used as input to the diffusion model. Subsequently, the denoising diffusion probabilistic model (DDPM) was trained as a convolution-based U-Net model. The data, generated in two-dimensions through the reverse process of diffusion, reshaped into one-dimension for

classification purposes. Diffusion's U-Net architecture (*Ronneberger, Fischer & Brox, 2015*) encompasses a contracting path, an expanding path, and a bridge. The contracting path employs convolution and pooling operations on input data to acquire context. In contrast, the expanding path utilizes features extracted from the contracting path to capture details, incorporating convolution and upsampling operations. Ultimately, the bridge facilitates the connection of features obtained from both the contracting and expanding paths, thereby enhancing overall performance.

Given the morphological similarity observed in certain classes of ECG data, it becomes imperative to generate data that effectively encapsulates the distinctive features of each class. Thus, this study applies classifier-free guidance (*Ho & Salimans, 2022*) as a means to generate data. Classifier-free guidance facilitates the training process of DDPM by incorporating class label information. The mechanism of classifier-free guidance is represented by Eq. (8).

$$\tilde{\epsilon}_\theta(\mathbf{z}_\lambda, \mathbf{c}) = (1 + \omega)\epsilon_\theta(\mathbf{z}_\lambda, \mathbf{c}) - \omega\epsilon_\theta(\mathbf{z}_\lambda) \tag{8}$$

In Eq. (8), $\epsilon_\theta(\mathbf{z}_\lambda, \mathbf{c})$ represents a conditional diffusion model with class labels as input and $\epsilon_\theta(\mathbf{z}_\lambda)$ means an unconditional diffusion model that trains without class labels and $\omega$ is the weight value. Consequently, the reverse process is acquired through the interpolation of the conditional diffusion model and the unconditional diffusion model. In summary, this study employs class labels as input and employs U-Net to learn the data distribution. Subsequently, data from classes S, V, and F are sampled based on the number of N class data. To evaluate the performance of the conditional diffusion model, the test data, which is 10% of the total dataset, was used to calculate the average cosine similarity per class. In this case, the cosine similarity was calculated using the actual noise and the noise predicted by the model. The obtained cosine similarity values are as follows: 0.989 for the S class, 0.992 for the V class, and 0.997 for the F class. These results indicate that the conditional diffusion model successfully learned the data distribution.

## IMPLEMENTATION

The process by which the model is trained to classify arrhythmias from ECG data is depicted in Fig. 3. The MIT-BIH dataset, characterized by the presence of baseline noise, so the preprocessing to eliminate this noise is required. Additionally, the dataset is comprised of 30 min of continuous data, necessitating segmentation into individual heartbeats. These segments are subsequently augmented, followed by the classification of arrhythmia classes. In this section, this research describes the data characteristics and preprocessing methods, and then describes the segmentation, augmentation, and classification methods based on how the heartbeats are segmented.

### ECG dataset & preprocessing

The data used in this study originates from the MIT-BIH arrhythmia dataset (*Goldberger et al., 2000*; https://physionet.org/content/mitdb/1.0.0/), comprising 30 min long ECG records extracted from 47 individuals monitored continuously over a 24 h. Each record was sampled at a frequency of 360 Hz and included both a modified limb lead II (MLII) and a modified

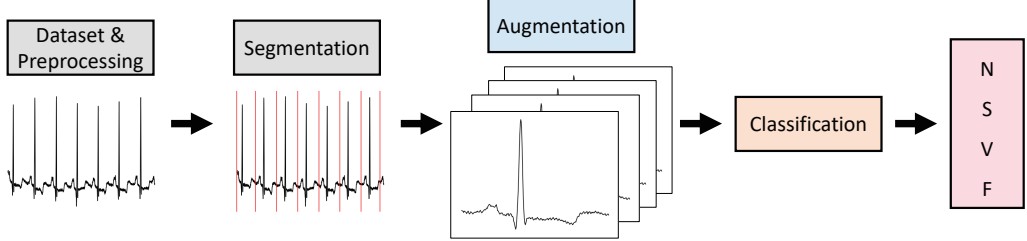

**Figure 3** Flow of arrhythmia classification models using ECG data.

lead V1 (occasionally V2 or V5, V4) measured using chest electrodes. For the purposes of analysis, only the MLII data was utilized. The R-peak and arrhythmia labels were obtained from Physionet and were curated by two cardiologists. Records 102, 104, 107, and 217 were excluded in accordance with AAMI standard (*Association for the Advancement of Medical Instrumentation, 1998*). Additionally, due to sparse data, the Q class data was excluded, and this study focused on classifying four distinct arrhythmia classes: N, S, V, and F.

When acquiring ECG signals, the MIT-BIH dataset often contains noise due to factors such as human movement or machine voltage fluctuations, necessitating preprocessing procedures (*Dasan & Panneerselvam, 2021*). In this study, the butterworth filter is applied in accordance with the approach detailed in *Qiu et al. (2021)* to mitigate machine voltage-induced noise effectively. Also, ECG signals may exhibit varying amplitudes among different subjects. To achieve stable and consistent learning, min-max normalization was utilized, where each sample's amplitude was rescaled to encompass the 0 to 1 range. Subsequently, all the data collected for this study performed the previously mentioned preprocessing procedures.

## Proposed ECG segmentation models

Among the diverse arrhythmias, premature contraction arrhythmias are distinguished by a narrow RRI within the ECG signal. Consequently, dividing the heartbeat into specified segment lengths may encompass patterns of heartbeats directly before or after the current heartbeat. Such segmentation serves as a potential feature for classifying premature contraction arrhythmias, yet it may pose challenges for learning algorithms. Therefore, this article compares an annotation-based segmentation model, which uses R-peaks provided in the MIT-BIH dataset, with an automated segmentation model that uses the Faster R-CNN deep learning framework.

### Annotation-based segmentation model

The structure of the model that performs annotation-based segmentation and classifies arrhythmias is shown in Fig. 4. In the MIT-BIH dataset, the R-peak label of each heartbeat is mapped. The study was divided into separate segments between 0.25 s before and 0.83 s after the mapped R-peak. The MIT-BIH dataset already provides arrhythmia labels for each R-peak, facilitating our use of this data for augmentation. To address imbalance, this research generated additional segment data for the imbalanced classes S, V, and F to

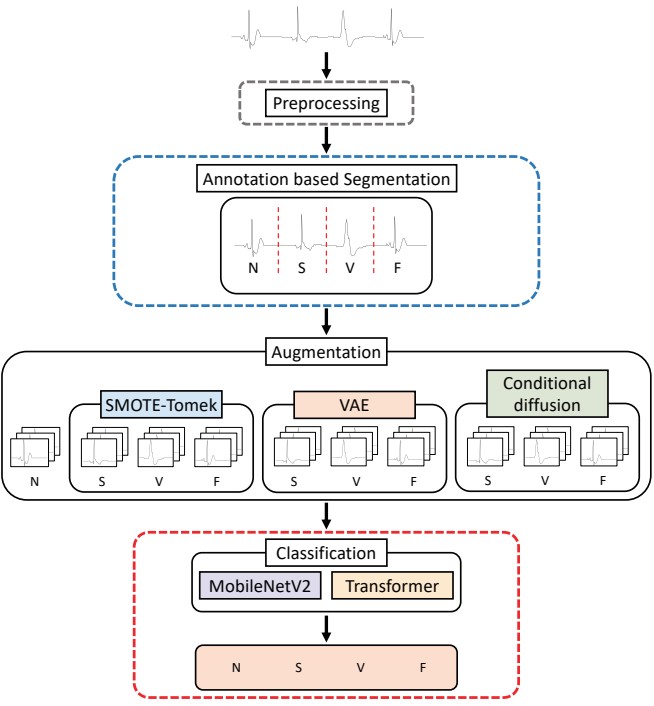

**Figure 4** **Structure of annotation-based segmentation model.**

equalize them with the N class using augmentation techniques such as SMOTE-Tomek, VAE, and conditional diffusion. This augmented data was then merged with the original dataset to create a comprehensive training dataset. Subsequently, the dataset was then trained to classify four arrhythmia classes using MobileNetV2 and the transformer models. The specifics of the arrhythmia classification process are further elaborated in classification section.

### Automated segmentation model

Figure 5 presents the configuration of the model designed for automated segmentation employing deep learning frameworks. This model was implemented the Faster R-CNN architecture, similar to that described in *Qiu et al. (2021)*, although certain parameters were adjusted to meet the constraints of the experimental setup. The preprocessing procedure adopted for this automated segmentation model is consistent with that of the annotation-based segmentation model.

The first step of the automated segmentation model is "Feature Extraction", which is a step to train the features of the classes to extract the feature weights of the model. The Faster R-CNN model facilitates object detection by leveraging a detailed feature map of the data. Consequently, for effective ECG classification, it is essential to derive a feature map that adequately represents the data's attributes. In this study, the "Feature Extraction" process, indicated by the green dotted line in Fig. 5, is designed to extract the model's feature weights for the purpose of training the characteristics of the arrhythmia classes N, S, V, and F. The

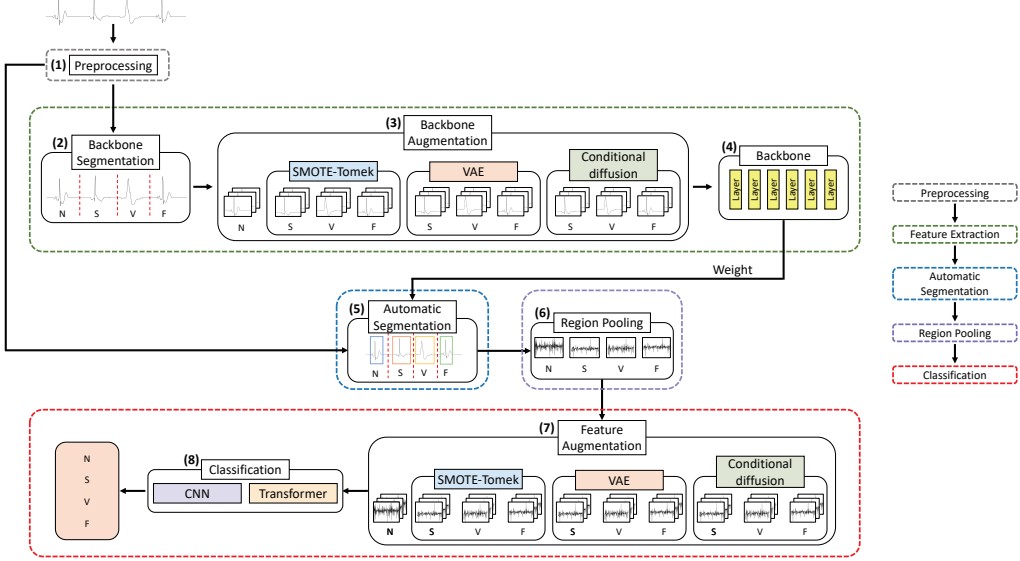

**Figure 5** Structure of automated segmentation model.

segmentation framework, forming the backbone of our approach, utilizes R-peaks on the preprocessed data to produce segments for each class. Since class imbalance is prevalent, Backbone Augmentation is performed to generate additional segments, using methods such as SMOTE-Tomek, VAE, and conditional diffusion to augment the representation of the minority classes. Fundamentally, as illustrated in Fig. 5, Backbone Augmentation is designed to increase the number of representations of features for imbalanced classes, thus promoting a more equitable training of features. After this augmentation, the backbone process uses MobileNetV2 to deepen the learning and refinement of each class's features. As a result, the "Feature Extraction" step is a collection of trained weight values, which is the basis for the next segmentation step.

The second step of the automated segmentation model is "Segmentation". The "Segmentation" process plays a crucial role in Faster R-CNN by predicting the regions where objects are located. This process leverages data from consecutive beats and labeled information to calculate a feature map through the application of a pretrained backbone. This procedure generates anchor boxes of different predefined sizes for predicting the regions containing objects. The next step is to execute a region proposal network (RPN), which is responsible for calculating the probability of object presence for each anchor box and adjusting the coordinates of the box using a bounding box regressor. The process can extract candidate regions using anchor boxes and object presence probabilities and bounding box regressors. This is the process of learning anchor boxes to better detect the location of objects. Since arrhythmias are caused by the heart beating irregularly and the length of each heartbeat is not constant, Faster R-CNN can effectively segment candidate regions. Using the feature map extracted by the Backbone network and the predicted candidate regions, select the anchor boxes with a high probability of object presence. This

process is called region of interest (ROI). However, since the ROIs are not uniform in size, training requires the next step: Region Pooling.

The third step is "Region Pooling", which equalizes the size of the data for training. Owing to the variability in segment sizes across different types of arrhythmias, the dimensions of the predicted ROI are not constant and it is important to standardize the size of the input data for effective classification of arrhythmia classes with deep learning. Consequently, "Region Pooling" is necessary to normalize the segments of varying sizes. "Region pooling" is the technique of transforming the feature map of an ROI into a fixed-size vector through max pooling. In our study, max pooling was employed on the ROIs within the feature map generated by the backbone process, ensuring uniformity in ROI sizes. In other words, the feature map of the ROI was segmented into small, fixed-size windows, with max pooling applied to each window. This approach effectively standardized the feature map dimensions while preserving only the most significant features from the ROIs of disparate sizes, thus maintaining a consistent feature map size.

The fourth step is the "Classification" step, which is the model that classifies arrhythmias into the red areas in Fig. 5. The outcome of "Region Pooling" is just dividing the preprocessed data into segments of uniform size. The MIT-BIH dataset is inherently imbalanced, exhibiting significant class imbalances. To address this issue, Feature Augmentation is conducted utilizing techniques such as SMOTE-Tomek, VAE, and conditional diffusion. Subsequently, a train dataset that integrates both the augmented and original data is employed in conjunction with CNN and the transformer models to classify the arrhythmia classes N, S, V, and F. Further details on the classification of arrhythmias are provided in classification section.

## ECG augmentation

The original in Fig. 6 illustrates the outcomes of augmented data corresponding to each class in the original dataset, representing the data employed for training. The generated waveforms for classes S, V, and F using augmentation techniques such as SMOTE-Tomek, VAE, and conditional diffusion are presented. In Fig. 6, it can be observed that, within the S class, all three augmentation techniques successfully preserved the morphological features while eliminating P-waves. Similarly, for class V, these techniques consistently yielded waveforms characterized by reduced S-wave. Moreover, in the case of the F class, the generated heartbeats displayed reduced amplitude in their T-waves across all augmentation techniques. However, it is important to note that the data generated by VAE exhibited noise at the onset of both the S and F classes. The limitation observed here pertains to the VAE augmentation model. Nonetheless, it is noteworthy that the noise pattern bears resemblance to the artifacts induced by motion during ECG measurements, suggesting the potential for training a noise-robust arrhythmia classification model. Furthermore, the data generated by the conditional diffusion model displayed lower T-wave amplitudes in comparison to the original data, which means that it generated more diverse data. Consequently, it is significant that all three augmentation techniques successfully maintained the morphological features of each class.

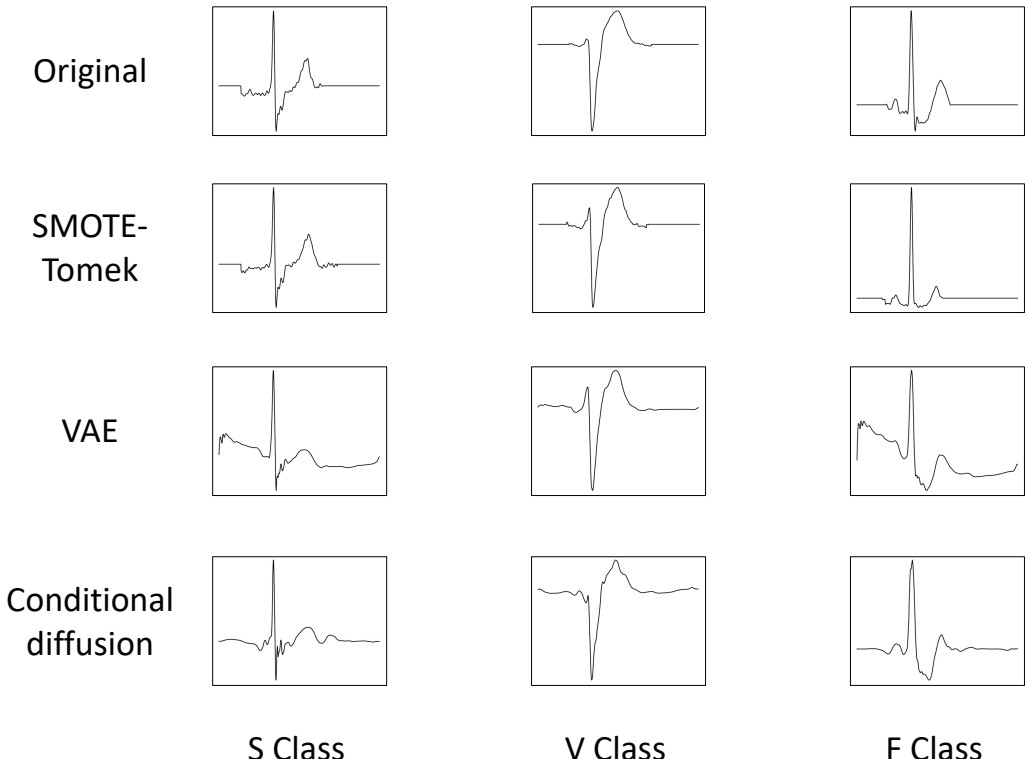

**Figure 6** ECG data generated using augmentation techniques for each classification.

## ECG classification model

In this study, the segmentation of the MIT-BIH arrhythmia dataset into individual heartbeats was carried out in the Augmentation section, encompassing both annotation-based segmentation and automated segmentation models. Following the segmentation process, segmented data were generated using SMOTE-Tomek, VAE, and conditional diffusion augmentation methods. Within the annotation-based segmentation model, the segments were categorized into four classes, namely N, S, V, and F, employing MobileNetV2 and the transformer models. Meanwhile, the automated segmentation model utilized CNN and the transformer models to classify the segments into four arrhythmia classes.

MobileNetV2 was used because it is a lightweight model that achieves high performance while using minimal memory. It also has residual layers, which makes training more stable. CNN is a model with proven performance that has been widely used in studies using time series data as well as image data. Therefore, CNN was used because it provides a baseline against which our findings can be compared to existing research. One of the main goals of this study is to experiment with and analyze the effectiveness of data augmentation techniques, and CNN is a suitable model to achieve this goal. The reason for using the transformer model as a classifier is that it has shown breakthrough performance in the field of language, a form of time series data. Moreover, it has the advantage of being parallelizable,

**Table 3  Proposed MIT-BIH classification models.**

| Models | Segmentation | Classification | Augmentation |
|---|---|---|---|
| Model 1 | Annotation-based | MobileNetV2 | – |
| Model 2 | | | SMOTE-Tomek |
| Model 3 | | | VAE |
| Model 4 | | | Conditional diffusion |
| Model 5 | | Transformer | – |
| Model 6 | | | SMOTE-Tomek |
| Model 7 | | | VAE |
| Model 8 | | | Conditional diffusion |
| Model 9 | Automated | CNN | – |
| Model 10 | | | SMOTE-Tomek |
| Model 11 | | | VAE |
| Model 12 | | | Conditional diffusion |
| Model 13 | | Transformer | – |
| Model 14 | | | SMOTE-Tomek |
| Model 15 | | | VAE |
| Model 16 | | | Conditional diffusion |

which allows it to process large amounts of data quickly. Therefore, a comparative analysis of the 16 models for arrhythmia classification is presented in Table 3.

Models 1–8 utilized the annotation-based segmentation technique, while Models 9–16 adopted the automated segmentation approach. Within Models 1–4, the segments obtained from annotation-based segmentation were classified using MobileNetV2, and Models 5–8 utilized the transformer for classification. On the other hand, Models 9–12, the segments obtained from automated segmentation were categorized using CNN, and Models 13–16 were classified using the transformer. The choice of using CNN as a classifier for Models 9–12 in the automated segmentation process was influenced by the need for a more lightweight model, given the constraints of the experimental environment.

Figure 7A shows the CNN and Fig. 7B visualizes the structure of the transformer classifier from Table 3. The CNN classifier took as input the features after region pooling. Features were extracted using one-dimension convolution with a kernel size of 3. In this case, stride was set to 1 to generate a detailed feature map, and the padding parameter was set to 1 to keep the size of the output the same as the input. Then, batch normalization layer and ELU activation function were added to stabilize the training. This process was repeated once more, followed by the addition of a dropout layer to prevent overfitting. Furthermore, convolution was used to extract deeper features from the data, and average pooling was employed to extract balanced features. Finally, a fully connected layer was used to classify the data into one of the four classes.

The input to the proposed the transformer classifier, input embedding, is the feature after region pooling. The input to the encoder is the input embedding added to the positional encoding. The model included a multi-head attention with four heads and a position-wise feed forward network, and this encoder block was repeated three times in total. After

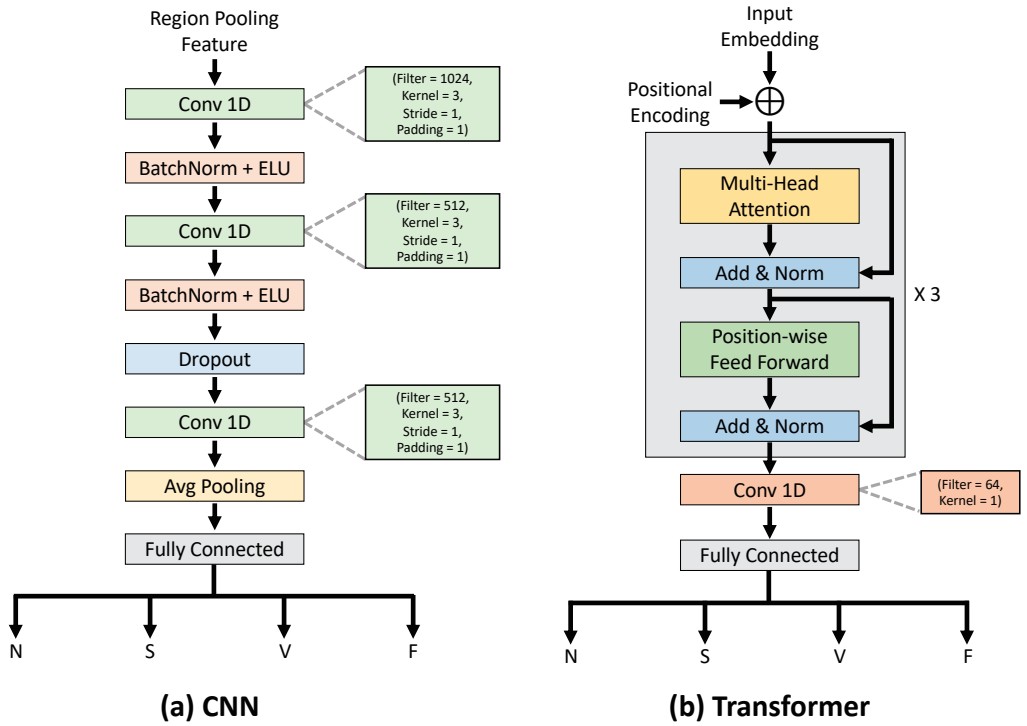

**Figure 7  The detailed structures of the CNN and transformer classifiers.**

applying one-dimension convolution to the encoder's output to extract detailed features, the model finally classified the data into one of the four classes using a fully connected layer.

# RESULT

In this research, the proposed models were implemented using PyTorch version 1.8.0 and dual NVIDIA GeForce RTX 3090 Ti GPUs. The Adam optimizer was used with a learning rate of 0.001 and a batch size of 240. All models were trained for 100 epochs. The MIT-BIH dataset was randomly split into 80% train data, 10% validation data, and 10% test data. Data augmentation was performed using only the train data, and the augmented data was combined with the original train data to construct the final train dataset for the experiments. Precision, sensitivity, F1 score, and accuracy were used to evaluate the performance of the arrhythmia classification models.

Table 4 presents a summary of the arrhythmia classification outcomes derived from methods that use annotation-based segmentation, as introduced within the models proposed in this research. In Table 4, the F1 scores for the minority classes S, V, and F were observed to be highest in Model 4. This evidence supports the conclusion that the conditional diffusion model is effective in mitigating class imbalance and enhancing the performance of arrhythmia classification. This is in the same context as articles (*Pandey & Janghel, 2019*; *Kuznetsov et al., 2021*; *Hairani, Anggrawan & Priyanto, 2023*;

**Table 4  Comparison of classification performance with annotation-based segmentation model.** The highest performance on each metric for each class is indicated in bold.

| Model \ Classification | N | | | S | | | V | | | F | | | |
|---|---|---|---|---|---|---|---|---|---|---|---|---|---|
| | PPV | SEN | F1 | PPV | SEN | F1 | PPV | SEN | F1 | PPV | SEN | F1 | ACC |
| Model 1 | 98.22 | 97.66 | 97.94 | 92.46 | 93.69 | 93.07 | 97.63 | 97.91 | 97.77 | 87.50 | 89.74 | 88.61 | 97.01 |
| Model 2 | 98.62 | 96.03 | 97.31 | 85.55 | **96.35** | 90.63 | 97.52 | 93.30 | 95.36 | 67.39 | **93.59** | 78.50 | 95.21 |
| Model 3 | 98.57 | 97.59 | 98.08 | 93.73 | 94.35 | 94.04 | 97.51 | 98.32 | 97.91 | 80.95 | 87.18 | 83.95 | 97.09 |
| Model 4 | **98.93** | 95.68 | **98.79** | 94.68 | 94.68 | **94.68** | **98.20** | 98.88 | **98.54** | 90.91 | 89.74 | **90.32** | **97.97** |
| Model 5 | 98.37 | **98.51** | 98.44 | **95.86** | 82.36 | 94.08 | 97.26 | **99.16** | 98.20 | **91.78** | 85.90 | 88.74 | 97.57 |
| Model 6 | 98.65 | 98.23 | 95.44 | 93.44 | 94.68 | 94.06 | 98.18 | 97.91 | 98.04 | 84.15 | 88.46 | 86.25 | 97.41 |
| Model 7 | 98.09 | 98.09 | 98.09 | 92.43 | 93.36 | 92.89 | 97.90 | 97.77 | 97.83 | 86.84 | 84.62 | 85.71 | 97.01 |
| Model 8 | 97.95 | 98.16 | 98.05 | 94.52 | 91.69 | 93.09 | 97.65 | 98.46 | 98.05 | 84.62 | 84.62 | 84.62 | 97.05 |

*Yamasaki et al., 2023*) that apply augmentation techniques to medical data to improve the performance of classification metrics. In Table 4, the average F1 scores for Models 1 through 4, which utilize MobileNetV2 as the classifier, were recorded at 94.35%, 90.45%, 93.5%, and 95.58%, respectively. This indicates that, in comparison to Model 1, which was trained on the original dataset, Model 4 exhibits a performance enhancement of 1.23%. The average F1 scores for Models 5 to 8, which utilize the transformer model as their classifier, were recorded at 94.87%, 94.2%, 93.63%, and 93.45%, respectively. While Model 5 registered the highest F1 score among Models 5 through 8, a comparative analysis of Models 1 through 8 revealed that Model 4 exhibited the best performance. Similarly, Model 4 also achieved the highest accuracy across all classes. This implies that Model 4 enhanced its classification efficacy due to the classification model's capability to assimilate the data generated through understanding the distribution of minority class data *via* conditional diffusion. Consequently, within the annotation-based segmentation models, Model 4 demonstrates superior performance in classifying with MobileNetV2, attributed to the learning from data augmented by conditional diffusion.

Table 5 summarizes the arrhythmia classification results of the automated segmentation model among the proposed models in this article. Model 10 represents that modifies certain parameters from the model presented in *Qiu et al. (2021)*. Initially, the average F1 scores for Models 9 to 12 were recorded at 82.9%, 26.86%, 61.57%, and 81.92%, respectively. Conversely, the average F1 scores for Models 13 to 16 were 90.16%, 90.66%, 88.06%, and 88.45%, respectively. Upon examination of these results, it is evident that Model 14 achieves the highest performance, particularly when trained on data augmented with SMOTE-Tomek and utilizing the transformer for classification.

In addition, this study aims to compare and analyze the models using accuracy metrics. Comparing the transformer-based classifiers Model 5-8 and Model 13-16, automated segmentation models showed superior accuracy over annotation-based segmentation models. This means that models that perform object detection and classification simultaneously have a positive impact on classification performance. *Qiu et al. (2021)* achieved 95.68% accuracy in arrhythmia class classification using the SMOTE-Tomek technique for data augmentation and the Faster R-CNN model. Model 12 proposed

**Table 5  Comparison of classification performance with automated segmentation model.** The highest performance on each metric for each class is indicated in bold.

| Model \ Classification | N | | | S | | | V | | | F | | | |
|---|---|---|---|---|---|---|---|---|---|---|---|---|---|
| | PPV | SEN | F1 | PPV | SEN | F1 | PPV | SEN | F1 | PPV | SEN | F1 | ACC |
| Model 9 | 99.03 | 99.29 | 99.16 | 89.25 | 82.16 | 85.56 | 94.83 | **96.02** | 95.42 | **87.32** | 36.47 | 51.45 | 98.25 |
| Model 10 | **100** | 1.64 | 3.22 | 81.07 | 52.07 | 63.42 | 93.69 | 25.10 | 39.59 | 0.61 | **100** | 1.20 | 6.59 |
| Model 11 | 99.80 | 76.87 | 86.84 | 12.34 | **95.78** | 21.86 | 93.28 | 94.99 | 94.13 | 29.10 | 85.71 | 43.44 | 79.19 |
| Model 12 | 99.05 | 98.05 | 98.55 | 59.15 | 88.55 | 70.92 | **99.23** | 87.99 | 93.28 | 58.77 | 72.52 | 64.92 | 96.94 |
| Model 13 | 99.06 | **99.56** | **99.31** | 89.89 | 82.34 | **85.95** | 98.25 | 95.25 | **96.73** | 81.65 | 75.88 | 78.66 | **98.62** |
| Model 14 | 99.06 | 99.46 | 99.26 | 89.18 | 82.55 | 85.74 | 97.50 | 95.85 | 96.67 | 85.16 | 77.19 | **80.98** | 98.57 |
| Model 15 | 98.82 | 99.53 | 99.17 | **91.38** | 74.14 | 81.86 | 96.01 | 96.01 | 96.01 | 80.89 | 70.27 | 75.21 | 98.27 |
| Model 16 | 98.89 | 99.46 | 99.17 | 86.50 | 81.39 | 83.87 | 97.15 | 94.54 | 95.83 | 81.94 | 69.01 | 74.92 | 98.37 |

in this study, utilizing conditional diffusion augmentation technique, achieved 98.07% accuracy in arrhythmia classification. This demonstrates that compared to the SMOTE-Tomek augmentation technique, conditional diffusion augmentation performs better in arrhythmia classification. The results of this study suggest that high-quality data augmentation significantly improves the accuracy of arrhythmia diagnosis and validate the superiority of the proposed approach.

Next, this research examines the confusion matrix of Model 4, which has the highest F1 score among the 16 models, to understand the problem of our proposed model. As shown in Table 6, Model 4 struggled with accurately classifying the N and S classes. The reason for this limitation is the morphological similarity of the two classes. It is judged that the model did not clearly train the features necessary to distinguish between the two classes.

## DISCUSSION

The performance of various augmentation techniques was evaluated to address the class imbalance. The findings from this evaluation were discussed to explore potential improvements in ECG classification performance.

Table 7 provides a comparative analysis of the performance between the proposed approach and existing ECG classification techniques. *Zhu et al. (2018)* utilized the Pan-Tompkins algorithm for QRS detection, followed by extracting morphological features through PCA and DTW. Using the SVM-RBF method, they classified four classes-N, S, V, and F. *Nurmaini et al. (2019)* employed a deep auto-encoder (DAE) for feature extraction and utilized a deep neural network (DNN) to classify 10 arrhythmia types. *Yu (2020)* utilized wavelet transform for detecting R-peak and RRI, and employed a CNN for the classification of four AAMI classes. *Rafi & Akthar (2021)* implemented a hybrid approach combining CNN and LSTM networks to enhance classification accuracy. *Shoughi & Dowlatshahi (2021)* applied SMOTE to augment the datasets for S and F classes, subsequently classifying five AAMI classes through CNN-BLSTM. *Aphale, John & Banerjee (2021)* applied Daubechies wavelet transform for feature extraction and employed SMOTE to address class imbalance, using ArrhyNet with a CNN to classify five AAMI classes. *Bhatia et al. (2022)* augmented the dataset using SMOTE and introduced a classification

**Table 6  Confusion matrix with suggested model which is composed of conditional diffusion augmentation, annotation-based segmentation and MobileNetV2 classification (Model 4).**

| Actual \ Predicted | N | S | V | F |
|---|---|---|---|---|
| N | 1392 | 14 | 2 | 3 |
| S | 10 | 285 | 6 | 0 |
| V | 3 | 1 | 708 | 4 |
| F | 2 | 1 | 5 | 70 |

model with CNN-BLSTM. Upon comparing the models that employ annotation-based segmentation, as detailed in Table 7, it becomes clear that Model 4, introduced in this study, exhibits superior performance relative to its counterparts. To analyze the F1 score of each class in Model 4, we compared the articles in Table 7 with those that applied the augmentation technique. Model 4 outperformed the S, V, and F classes from *Shoughi & Dowlatshahi (2021)* by 7.18%, 2.6%, and 4.96%, respectively. Furthermore, compared to *Aphale, John & Banerjee (2021)*, it achieved superior results with improvements of 8.68%, 6.54%, and 2.32%. Finally, compared to *Bhatia et al. (2022)*, it achieved higher F1 scores of 6.77%, 3.07%, and 13.79%.

Through this result comparison, the proposed model showed a significant improvement in performance, especially in minority classification. This suggests that the conditional diffusion technique more accurately reflects the distribution of the original data and generates data that better preserves the features of minority classes. This characteristic enables the model to better train minority classes, which ultimately leads to a significant improvement in F1 score. Therefore, this study is significant not only for achieving a high F1 score but also for paving the way for the development of more accurate and reliable arrhythmia diagnosis systems. These results indicate that the approach proposed in this study can be effectively utilized in future medical data analysis and diagnosis system development.

Subsequently, the models that classified arrhythmias with the automated segmentation method were compared. Tables 4 and 5 shows that Model 14 performed the best among the proposed methods. In previous research, *Ji, Zhang & Xiao (2019)* utilized the Faster R-CNN framework to identify five specific classes (Normal, LBBB, RBBB, PVC, and FVN) which represent a subset of the AAMI class. *Hwang et al. (2020)* employed the YOLO algorithm to categorize arrhythmias into the classes Normal, LBBB, RBBB, S, and V. In summary, both models were classified by the class of MIT-BIH symbol, which is a subclassification system of N, S, V, and F, rather than the classification classes of AAMI. However, in this study, all data were extensively processed in accordance with the AAMI class, as opposed to the subset of MIT-BIH classification symbols, and were classified through automated segmentation. For the arrhythmia classification model to be used as a supplementary model to deal with the full range of arrhythmia data, it is crucial that it encompasses a broad range of data, including all diverse classes, rather than focusing on a model that categorizes a small subset of detailed classes.

The limitations of our study are as follows. Although most classes achieved high F1 scores in an imbalanced label environment, the classification performance of the F class, which

**Table 7 The correspondence between AAMI classes and MIT-BIH arrhythmia classes.**

| Segmentation | Reference | Augmentation | Class | | Classifier | F1 score (%) |
|---|---|---|---|---|---|---|
| | | | **AAMI class** | **MIT-BIH class** | | |
| Annotation-based model | *Zhu et al. (2018)* | - | N, S, V, F | - | SVM-RBF | 91.14 |
| | *Nurmaini et al. (2019)* | - | - | A, L, N, P, R, V, f, F, !, j | DAE+DNN | 91.80 |
| | *Yu (2020)* | - | N, S, V, F | - | CNN | 88.21 |
| | *Rafi & Akthar (2021)* | - | N, S, V, F, Q | - | CNN+LSTM | 87.40 |
| | *Shoughi & Dowlatshahi (2021)* | SMOTE | N, S, V, F, Q | - | CNN-BLSTM | 93.45 |
| | *Aphale, John & Banerjee (2021)* | SMOTE | N, S, V, F, Q | - | ArrhyNet | 93.00 |
| | *Bhatia et al., (2022)* | SMOTE | N, S, V, F, Q | - | CNN-BLSTM | 91.67 |
| | Model 4 | Conditional Diffusion | N, S, V, F | - | MobileNetV2 | **95.58** |
| Automated model | *Ji, Zhang & Xiao (2019)* | - | - | Normal, L, R, V, F | Faster R-CNN | 98.04 |
| | *Hwang et al. (2020)* | - | - | Normal, L, R, S, V | YOLO | 96.00 |
| | Model 14 | SMOTE-Tomek | N, S, V, F | - | Faster R-CNN-Transformer | 90.66 |

has a smaller amount of data, remained relatively low. While the conditional diffusion technique contributes to maximizing arrhythmia classification performance, there is a possibility that the augmented data may be misclassified into other classes, which can cause confusion during the model training process. Furthermore, these augmentation techniques may excessively reflect the patterns of the original data, potentially leading to model overfitting. Although the proposed model demonstrated high performance on the dataset used in this study, further research is needed to determine if it can lead to the same performance improvement on other datasets.

## CONCLUSION

This study introduces and assesses a model designed to accurately classify arrhythmias using the imbalanced MIT-BIH arrhythmia dataset. The proposed arrhythmia classification model consists of four processes: preprocessing, segmentation, augmentation, and classification. The preprocessing step involves removing noise and normalizing the waveform, while the segmentation process refers to dividing the data into units representing a single heartbeat. In this procedure, annotation-based segmentation and automated segmentation models are presented. The annotation-based segmentation model employs a method that segments data using R-peak annotations already provided in the dataset. In contrast, the automated segmentation model utilizes the Faster R-CNN, a deep learning framework, for performing segmentation tasks. The automated segmentation model implements the augmentation of imbalanced data across different classes to achieve more robust segmentation outcomes. Based on the augmented data, model trained features for each class. These feature maps were then utilized to generate features of uniform size for region pooling training, because the segmentation unit size is different when automatically segmenting. After heartbeat segmentation through either annotation-based or automated segmentation methods, augmentation techniques were applied to enhance the classification performance of minority classes. Augmentation techniques, including SMOTE-Tomek, VAE and conditional diffusion, have been shown to improve classification performance

by increasing the volume of training data available for minority classes. In the final step of classification, the annotation-based segmentation model employed MobileNetV2 and the transformer as classifiers, while the automated segmentation model utilized CNN and the transformer for classification purposes. Among the 16 methods proposed, the annotation-based segmentation model demonstrated superior classification performance when utilizing MobileNetV2. This enhanced performance was attributed to MobileNetV2's learning from data generated by the conditional diffusion model, which was based on annotation-based segmentation.

The objective of this study is to improve the efficacy of arrhythmia classification models by integrating all classifications for arrhythmias as specified in the AAMI classification. Unlike previous research, which typically focused on classification models for certain MIT-BIH symbols, this study encompasses all classification types across all AAMI categories. To evaluate our performance against previous studies, we examined both the automated segmentation method and the annotation-based segmentation method. In the proposed automated segmentation method, the transformer model, trained on data augmented with SMOTE-Tomek, exhibited the highest classification performance for datasets that included various arrhythmia patterns. As a result of comparing the classification performance of annotation-based segmentation, MobileNetV2, which learned the data generated by the conditional diffusion model, had the best classification performance. This study demonstrates that the conditional diffusion augmentation technique can significantly improve the performance of arrhythmia classification models. In particular, the improvement of F1 score in the minority class indicates that this technique can effectively address the data imbalance problem. These achievements can be utilized as an important basis for the development of various medical data analysis and diagnosis systems in the future. Future research should conduct additional validation on various environments and datasets to increase the generalizability of the proposed model. Through this, we hope to develop more sophisticated and reliable medical data analysis models, ultimately contributing to the improvement of patient diagnosis and treatment quality.

### Funding
This research was supported by a National Research Foundation of Korea (NRF) grant funded by the Korean government (MSIT) (NRF-2022R1F1A1061476). The funders had no role in study design, data collection and analysis, decision to publish, or preparation of the manuscript.

### Grant Disclosures
The following grant information was disclosed by the authors:
National Research Foundation of Korea (NRF) grant funded by the Korean government (MSIT): NRF-2022R1F1A1061476.

### Competing Interests
The authors declare there are no competing interests.

## Author Contributions

- Jinhee Kwak conceived and designed the experiments, performed the experiments, performed the computation work, prepared figures and/or tables, authored or reviewed drafts of the article, and approved the final draft.
- Jaehee Jung conceived and designed the experiments, analyzed the data, performed the computation work, authored or reviewed drafts of the article, and approved the final draft.

## Data Availability

The data is available at MIT-BIH Arrhythmia Database:

https://physionet.org/content/mitdb/1.0.0;

https://doi.org/10.13026/C2F305.

## Supplemental Information

Supplemental information for this article can be found online at http://dx.doi.org/10.7717/peerj-cs.2299#supplemental-information.

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
