# Peer review of "Classification of imbalanced ECGs through segmentation models and augmented by conditional diffusion model"

_PeerJ Computer Science, doi:10.7717/peerj-cs.2299_

## Round 0.1 · original submission · Major Revisions

All comments from the expert reviewers must be considered to revise the manuscript

·

Basic reporting

Line 315: space before the period. i.e. "mapped.The"

Experimental design

No comment

Validity of the findings

no comment

Additional comments

Overall, no major revisions required. All the topics and details are captured in the great manner. I'd recomend breaking the paragraph down into smaller chunks for easier readability. Would try to reduce the content and remove some repeatitive aspects. Overall, great work.

Reviewer 2 ·

Basic reporting

All comments have been added in detail to the last section.

Experimental design

All comments have been added in detail to the last section.

Validity of the findings

All comments have been added in detail to the last section.

Additional comments

Review Report for PeerJ Computer Science
(Classification of imbalanced ECGs through segmentation models and augmented by conditional diffusion model)

1. Within the scope of the study, segmentation and arrhythmia classification operations were carried out with deep learning on the Electrocardiograms (ECG) signal dataset.

2. In the Introduction section; What is arrhythmia and its importance, the correspondence between the classes of The Association for the Advancement of Medical Instrumentation (AAMI) standard and the Massachusetts Institute of Technology-Boston's Beth Israel Hospital (MIT-BIH) arrhythmia database classes, deep learning and machine learning in the literature Important arrhythmia classification studies have been mentioned sufficiently. It is recommended to add the main contributions to the literature in clearer form at the end of this section, in order to more clearly emphasize the difference of the study from the literature.

3. In the study, the open source MIT-BIH dataset, which is frequently preferred for classification processes in ECG signals, was used. To eliminate dataset imbalance, Synthetic Minority Oversampling Technique, Tomek, Conditional Diffusion and Variational Autoencoder were preferred for the data augmentation stage. Instead of using the raw data in its original form, it is very valuable to use data augmentation to eliminate imbalance. Although there are many different methods that can be used for augmentation at this stage, it should be stated more clearly why these three augmentation methods are preferred and their superiority over others.

4. It is stated in Figure-5 that Faster R-CNN was used in the automatic segmentation phase. Although this section is explained sufficiently, although there are many different deep learning-based models that can be used in the segmentation stage, please explain more clearly the reason and advantage of using this model.

5. In the classification stage, sufficient applications have been carried out in many models with no augmentation or different augmentation types, including both annotation-based and automated segmentation. In this section, it is stated that Mobilenet, CNN and Transformer deep learning models are used as classifiers. Although there are many different deep learning-based classifiers in the literature, it should be stated more clearly why they are preferred. It has been stated that CNN and Transformer are used as classifiers, but since these are general nomenclature, it is recommended to add clearer information about the structure of the model details.

6. In the Evaluation metric section, there are some metrics (such as precision, recall, f1 score) for classification. However, in order to fully analyze the results, it is recommended to add missing metrics (such as accuracy-epoch graph, loss-epoch graph, confusion matrix, ROC curve, AUC score).

7. Information such as the framework/toolbox and hardware used during the implementation phase should be added in detail. In addition, information about the hyperparameters preferred for the deep learning models used within the scope of the study should be added in detail.

As a result, although the study is at a certain level in terms of both subject and originality, attention should be paid to the parts mentioned above.

Reviewer 3 ·

Basic reporting

Pros:

The manuscript is clearly written and provides a comprehensive background on ECG classification and the challenges posed by imbalanced datasets.
The introduction and background sections effectively establish the context and importance of the study, referencing relevant and recent literature.
Figures and tables are relevant, high quality, and well-labeled, enhancing the understanding of the study's results.
The raw data is supplied in accordance with the journal's policies, ensuring transparency and reproducibility.
Cons:

The abstract could be more concise, focusing more on the key findings and their implications rather than detailed background information.
Some sections could benefit from additional citations to strengthen the discussion, particularly in the methods and results sections. Including references to studies that have employed similar augmentation techniques in medical data classification would provide a stronger foundation (e.g., Chawla et al., 2002; Pandey & Janghel, 2019).

Experimental design

Pros:

The research question is well-defined, relevant, and meaningful, addressing the critical issue of class imbalance in ECG data.
The study employs rigorous and advanced data augmentation techniques (SMOTE-Tomek, VAE, and conditional diffusion) to address class imbalance, which is a significant contribution to the field.
The methods are described in sufficient detail to allow replication, including the specifics of the augmentation techniques and the classification models used.
Cons:

The rationale for choosing the specific augmentation techniques could be better justified. While SMOTE-Tomek and VAE are well-known, the novelty and advantages of using conditional diffusion could be highlighted more explicitly.
The manuscript would benefit from a clearer explanation of the segmentation process and how it impacts the overall classification performance. Comparing this with other segmentation techniques used in similar studies would provide better context (e.g., Xu et al., 2020; Qiu et al., 2021).

Validity of the findings

Pros:

The findings are robust, statistically sound, and well-supported by the data. The use of multiple evaluation metrics (precision, sensitivity, F1 score) provides a comprehensive assessment of model performance.
The study demonstrates that the proposed augmentation techniques significantly improve classification performance, particularly for minority classes, which is a crucial finding for the field.
Cons:

The discussion could be enhanced by including a more critical analysis of the limitations of the study, such as the potential overfitting with augmented data and the generalizability of the findings to other datasets.
Comparing the results with those of other studies in the field would help validate the conclusions. For example, discussing how the improvements in F1 scores compare to those achieved by other augmentation techniques in similar studies (e.g., Anis & Sharma, 2022; Rana & Kim, 2019) would provide valuable context.

Additional comments

Pros:

The manuscript addresses a significant challenge in ECG classification, offering valuable insights and practical solutions for handling imbalanced data.
The use of advanced data augmentation techniques and a comparative analysis of segmentation models are strong points of the study.
The conclusions are well-supported by the experimental results and provide clear directions for future research.
Cons:

The manuscript could benefit from a more detailed discussion of potential confounding factors and the implications of the findings for clinical practice.
Some minor language edits could improve readability and reduce redundancy, particularly in the introduction and discussion sections.

---

## Round 0.2 · accepted · Accept

It was good to see the revision and encouraging remarks from the expert reviewer

Reviewer 2 ·

Basic reporting

All comments have been added in detail to the last section.

Experimental design

All comments have been added in detail to the last section.

Validity of the findings

All comments have been added in detail to the last section.

Additional comments

Review Report for PeerJ Computer Science
(Classification of imbalanced ECGs through segmentation models and augmented by conditional diffusion model)

Thanks for the revision. The answers given, the changes made and the new version of the paper are sufficient. For this reason, I recommend that the paper be accepted. I wish the authors success in their future studies. Kind regards.